# Interaction Effects of DRD2 Genetic Polymorphism and Interpersonal Stress on Problematic Gaming in College Students

**DOI:** 10.3390/genes13030449

**Published:** 2022-02-28

**Authors:** Esther Kim, Dojin Lee, KyuMi Do, Jueun Kim

**Affiliations:** Department of Psychology, Chungnam National University, Daejeon 34134, Korea; esther170@naver.com (E.K.); ldj370@naver.com (D.L.); cherish1160@naver.com (K.D.)

**Keywords:** problematic gaming, *DRD2 C957T*, *DRD2 Taq1*, interpersonal stress, avoidant coping

## Abstract

Problematic gaming has become a public concern, influenced both by genetic factors and stressful environments. Studies have reported the effects of dopamine-related genes and interpersonal stressors on problematic gaming, but gene and environment interaction (G × E) studies have not been conducted. In this study, we investigated the interaction effects of dopamine receptor D2 (DRD2) polymorphisms and interpersonal stress on problematic gaming and the mediating effect of avoidant coping to reveal the mechanism of the G × E process. We recruited 168 college students (mean age = 22; male 63.1%) and genotyped their *DRD2 C957T* (rs6277) and *Taq1* (rs1800497) polymorphisms. The results of the mediated moderation analysis showed that, when experiencing interpersonal stressors, individuals with both the *C957T* T allele and the *Taq1* A1 allele showed more elevated problematic gaming scores than non-carriers. Moreover, the interaction effect of the combined DRD2 polymorphisms and interpersonal stress was significantly mediated by avoidant coping. These findings suggest that the influence of interpersonal stress on problematic gaming can be changed as a function of DRD2 genotypes, which may be because of the avoidant coping styles of *C957T* T allele and *Taq1* A1 allele carriers in response to stress.

## 1. Introduction

Opinions diverge as to whether excessive gaming is a disease, and discussions on definitions and diagnosis continue [1,2]. Despite the ongoing debate, the American Psychiatric Association included internet gaming disorder to in its *Diagnostic and Statistical Manual of Mental Disorders, Fifth Edition,* in 2013 as an area that needs further study [3]. The World Health Organization listed gaming disorder in the *11th International Classification of Diseases* (ICD-11) in 2019. The ICD-11 defined gaming disorder as “impaired control over gaming, increasing priority given to gaming over other activities to the extent that gaming takes precedence over other interests and daily activities, and continuation or escalation of gaming despite the occurrence of negative consequences” [4]. According to a meta-analysis on problematic gaming covering the years 2009–2019, the international prevalence of problematic gaming was about 2.09% [5]. Individuals who are excessively involved in gaming are likely to experience negative moods, such as irritability [6], poor physical health, and reduced sleep quality [7], as well as poor performance at work and school [8,9].

Problematic gaming is a behavior driven by the complicated interplay among biological, psychological, and social factors. An individual’s genetic makeup can be one significant factor, and among specific genes potentially associated with problematic gaming, dopamine receptor D2 (DRD2) polymorphisms can be promising candidate genes; functional deficits related to dopamine transmission have been strongly suggested as major factors in the development of addiction [10]. The *DRD2 C957T* polymorphism is located on chromosome 11, and DRD2 TaqIA is located adjacent to the end codon of DRD2 and adjacent to the ankyrin repeat and kinase domain containing 1 [11]. The T allele of *DRD2 C957T* has been associated with a decreased mRNA translation efficiency at the D2 receptor, resulting in impaired D2 receptor function [12]. Moreover, the A1 allele of *DRD2 Taq1* affects D2 receptor availability by decreasing the density of dopamine receptors [13,14].

There are a few studies that have reported *DRD2 Taq1* as a candidate gene for problematic gaming or excessive internet use [15,16]. The *DRD2 Taq1* A1 allele frequency was significantly higher in the problematic gamer group than with the non-problematic gamers, and problematic gamers with Taq 1 allele showed a markedly high tendency to respond to signals of rewards [15]. Similarly, the *DRD2 Taq1* A1 allele frequency was significantly higher in excessive internet users than in non-excessive internet users [16]. Regarding *DRD2 C957T*, there is no study on problematic gaming, but there are several studies that have reported the *DRD2 C957T* gene as a candidate gene for alcohol dependence [17,18,19].

Problematic gaming can be influenced by genetic factors, as well as by interpersonally stressful environments. When experiencing negative interpersonal situations, the severity of problematic gaming may increase because gaming becomes a strategy to avoid stress. Individuals experiencing interpersonal problems may satisfy basic psychological needs, such as belonging and autonomy, through gaming [20]. Some studies on problematic gaming have revealed that interpersonal stress (including conflicts with peers, lack of social support, or being bullied) can be a risk factor for problematic gaming. A longitudinal study with Dutch adolescents (*n* = 354) [21] showed that adolescents who have more difficulty maintaining close friendships with their peers reported increased severity of later problematic-gaming symptoms. Moreover, another large longitudinal study of Chinese adolescents (*n* = 2666) [22] reported that experiences of peer bullying or cyberbullying predicted later problematic gaming.

However, not all individuals who have experienced interpersonal stress become problematic gamers. Individuals with reduced dopamine-receptor function may be more involved in gaming, particularly when they are stressed because of interpersonal problems. It may be because those with reduced dopamine-receptor function are more sensitive to stressful situations and tend to cope with stress through gaming that provides stimulation and pleasure more than those with average levels of dopamine-receptor functions. Researchers have studied the interaction effects between DRD2 and stressful environments on problematic alcohol use. When experiencing many negative life events in the past 12 months, *DRD2 Taq1* A1 allele carriers showed higher levels of alcohol dependence than non-carriers [23]. Similarly, when experiencing higher levels of occupational or economic stress, *DRD2 Taq1* A1A1 carriers showed higher alcoholism scores than non-carriers [24]. As with previous studies, the effects of interpersonal stress on problematic gaming may differ as a function of DRD2 polymorphisms, and the interaction effects between DRD2 polymorphism and interpersonal stress need to be studied.

In any further examination of the possible reasons the interaction between DRD2 polymorphism and interpersonal stress may influence problematic gaming, it is necessary to examine a maladaptive coping style (avoidant coping) as a mediating factor. Avoidant coping refers to handling a problem by attempting to push away the cognitive, emotional, and behavioral aspects of a stressor [25,26]. Escapism is closely related to avoidant coping and refers to avoiding real-life problems by engaging in online activities [27,28]. The reason why this escapism is important in problematic gaming is that games (especially Massively Multiplayer Online Role Playing Games) allow users to escape from reality to “another world” [29]. When individuals experience negative life events and high levels of stress, they are likely to be motivated to escape to a gaming world to satisfy unmet needs or to alleviate unpleasant moods [30]. Individuals who respond in a consistently avoidant way have been found to be vulnerable to problematic gaming [31] and other addictive behaviors, such as binge drinking, excessive eating, and smoking [32,33,34]. Moreover, individuals who had the *DRD2 Taq1* A1 allele and experienced extreme emotional stress were likely to choose more avoidant coping methods than non-carriers, thus increasing the severity of alcohol abuse [35]. 

In this study, we examined the interaction effects between DRD2 polymorphisms (*DRD2 C957T* and *DRD2 Taq1*) and interpersonal stressors on problematic gaming among college students. We also investigated whether avoidant coping functions is a mediating variable that explains the mechanisms that cause the GxE effects on problematic gaming to manifest. We hypothesized that individuals who have the *DRD2 C957T* T allele and/or the *DRD2 Taq1* A1 allele would show more problematic gaming behaviors than non-carriers when experiencing interpersonal stressors. We also expected that the DRD2 risk allele carriers would be more likely to use avoidant coping in interpersonally stressful situations, which increased problematic gaming.

## 2. Materials and Methods

### 2.1. Participants

We recruited participants through university bulletin boards, social media, game-related websites, and local advertisements. Korean college students who reported spending an average of 1 or more hours a day playing games during the past 3 months were recruited. These screening criteria were based on a previous study reporting that playing for more than 1 h a day increases the risk of problematic gaming [36]. We recruited a total of 168 individuals: 106 participants were males (63.1%), and 62 were females (36.9%). The average age of the participants was 22 years (SD = 2.35, age range: 19 to 33 years old).

### 2.2. Measures

*Problematic gaming*. We used the 27-item Korean version of the Internet Gaming Disorder Scale to measure problematic gaming [37,38]. We rated each item on a 6-point Likert scale, ranging from 0 (not at all) to 5 (always). This scale has reliable convergence and concurrent validities [37]. The subscales included preoccupation, tolerance, withdrawal, persistence, negative consequences, deception, displacement, and conflict. In our study, the Cronbach’s alpha of the total items was 0.93. We used a sum of all items, with higher scores indicating a higher level of problematic gaming.

*Interpersonal stress*. We used the 5-item interpersonal stress subscale of Life Stress Scale for college students [39]. We rated each item on a 4-point Likert scale, ranging from 0 (not at all) to 3 (often). The scale items were “I was bullied by my friends”, “I was rejected by my friends”, “My friends ignored me”, “I was rude to my friends”, and “I couldn’t make friends I liked”. In our study, the Cronbach’s alpha of the items was 0.81. We used a sum of all five items, with higher scores indicating a higher level of interpersonal stress.

*Avoidant coping*. We used the 6-item avoidant coping subscale of the Ways of Coping Checklist [40,41]. We rated each item on a 4-point Likert scale, ranging from 1 (not used) to 4 (used a great deal). The scale items included “I tried to forget the whole thing”, “I acted as if nothing had happened”, and “I got mad at the other people or things”. In our study, the Cronbach’s alpha of the items was 0.78. We used a sum of all six items, with higher scores indicating a higher level of avoidant coping.

*Controlling variables*. Participants’ sex [42], impulsivity [43], anxiety [43], depression [42], and attention deficit hyperactivity disorder (ADHD) symptoms [44] were significantly correlated with problematic gaming, so we included these variables as covariates in all analyses to statistically control for the confounding effects. We measured participants’ self-reported sex (female = 0, male = 1). We used the 20-item UPPS-P Impulsive Behavior Scale to measure impulsivity [45] and the 7-item Generalized Anxiety Disorder Scale (GAD-7) to measure anxiety [46]. We used the 9-item Patient Health Questionnaire-9 (PHQ-9) to measure depression [47] and the 6-item Part A of the ADHD Self-Report Scale (ASRS) to measure ADHD symptoms [48]. In our study, the Cronbach’s alphas of the UPPS-P, GAD-7, PHQ-9, and ASRS were 0.82, 0.92, 0.84, and 0.62, respectively.

### 2.3. Genotyping

*DRD2 C957T* (rs6277, F: TGT GGT GTT TGC AGG AGT CT, R: CCT GCA GCC ATG GTT AGG AA) and *DRD2 Taq1* (rs1800497, F: AGG TAC ATC GTT ATG GCT TGG, R: ATA TTT GTG CAG TGC TGG GC) were initially extracted from saliva samples as DNA, using an AccuPrep® Genomic DNA Extraction Kit and amplified by using an AccuPower® ProFi Taq PCR Premix. The reaction occurred under the following conditions: 5 min of initial denaturation at 95 °C, followed by 35 cycles of denaturation at 95 °C for 30 s, 55 °C for 30 s, and 72 °C for 30 s, with a final elongation at 72 °C for 5 min. After amplification, all PCR reactions were confirmed with an agarose gel, using an AccuPrep® PCR/Gel purification Kit. Sequencing was performed by using a BigDye Terminator v3.1 sequencing kit (Thermo Fisher Scientific, Waltham, MA, USA).

To analyze two GxE interaction models, we recorded the T allele of *DRD2 C957T* and the A1 allele of *DRD2 Taq1* as risk alleles based on previous studies [12,13]. For the *DRD2 C957T* x interpersonal stress model, we coded T allele carriers as “1” and non-carriers as “0”. For the *DRD2 Taq1* x interpersonal stress model, we coded A1 allele carriers as “1” and non-carriers as “0”. In our study, the T allele for *C957T* polymorphism was considered a risky allele with more neurobiological evidence, as the T allele reduces the mRNA translation efficiency and attenuates the dopamine-induced upregulation of DRD2 expression [12,13]. However, because there are some conflicting findings on the risk alleles of the *DRD2 C957T* polymorphism [18,19], we conducted sensitivity analyses using other genotype categorizations (e.g., identifying the *C957T* C allele as a risk-conferring allele).

To analyze the GxGxE (Gene x Gene x Environment) interaction model, we recorded the combined genotypes of both the *C957T* polymorphism and the Taq1 polymorphism for each individual; if a person had both a T allele for *C957T* and an A1 allele for *Taq1* polymorphisms, their genotype was recorded as a T/A1 combined genotype and categorized as a “risk” genotype (we coded them as “1”). If a person had only one of the alleles (T or A1) or did not have either T or A1, their genotype was categorized as a “nonrisk” genotype (we coded them as “0”). This combined categorization was based on previous studies reporting that *DRD2 C957T* and *DRD2 Taq1* were located at a distance of 10 kb within chromosome 11 and had similar functions in determining the binding potential of the D2 receptor [49,50]. In some studies, researchers have examined the genetic risk binding of *DRD2 C957T* and *DRD2 Taq1* polymorphisms because of the possible interactions between them and have found individuals with the two risky alleles to be susceptible to alcohol [19] and nicotine dependence [51]. 

### 2.4. Statistical Analysis

We used SPSS 24.0 to conduct descriptive and bivariate correlation analysis and used Mplus 8.1 to conduct the mediated moderation and multigroup analysis. To address the non-normality of interpersonal stress, we used the Maximum Likelihood Estimation (MLE). The MLE method finds the parameter to be robust to non-normality because it finds the parameter most consistent with the sampling value based on the values sampled from a random variable [52].

We used Pearson’s correlation estimator to analyze continuous variables, and Spearman’s correlation estimator to analyze categorical variables. Regarding the mediated moderation model, we used the step-by-step approach that Baron and Kenny (1986) [53] and Muller and his colleagues (2005) [54] recommended. In Step One, we checked whether the interaction between genetic factors and interpersonal stress had a significant effect on problematic gaming. When the interaction path model showed a significant GxE effect, we conducted multigroup analysis as a post hoc test to estimate the simple effects of interpersonal stress on problematic gaming among DRD2 risk allele carriers and non-carriers. In Step Two, in the case of models in which the moderating effect was significant, we added an avoidant coping strategy as a mediator. To test the statistical significance of the indirect effects and verify whether the conditional indirect effect corresponded to the level of the mediating effects, we used a bootstrapping (extracting 10,000 repetitions) procedure [55].

## 3. Results

### 3.1. Descriptive and Correlation Analysis 

The descriptive and correlation results of study variables appear in Table 1. Interpersonal stress, avoidant coping, impulsivity, depression, anxiety, and ADHD symptoms were positively associated with problematic gaming (*rs* = 0.29 to 0.57, *p* < 0.001). In our data, male sex was negatively associated with problematic gaming, avoidant coping, impulsivity, depression, anxiety, and ADHD (*rs* = −0.36 to −0.15, *p* < 0.001 to 0.02). As presented in Table 2, there were no differences in the study variables between *DRD2 Taq1* risk allele carriers and non-carriers. Carriers of the T allele of DRD2 957T showed lower interpersonal stress, anxiety, and ADHD symptoms than non-carriers, with *ts* (18.48 to 85.86) = −4.01 to −2.13, and *p* = 0.001 to 0.047. Carriers of the T allele of *DRD2 C957T* and the A1 allele of *DRD2 Taq1* showed lower anxiety scores than non-carriers, with *t* (18.85) = −2.52, and *p* = 0.02. 

The distribution of the *DRD2 C957T* genotype in our participants was CC (*n* = 154, 91.7%), CT (*n* = 14, 8.3%), and TT (*n* = 0, 0.0%). The distribution of the *DRD2 Taq1* genotypes was A1A1 (*n* = 29, 17.3%), A1A2 (*n* = 90, 53.6%), and A2A2 (*n* = 49, 29.2%). Regarding the allele frequencies, the *DRD2 C957T* T allele was 14 (8.3%), the C allele was 322 (91.7%), the *DRD2 Taq1* A1 allele was 148 (44.1%), and the A2 allele was 188 (56.0%). This frequency was similar to the reported DRD2 allele frequency in a large public data source at the National Center for Biotechnology Information (NCBI). According to NCBI, the general frequency of the T allele of *DRD2 C957T* is 6.0% in Koreans, and the general frequency of the A1 allele of *DRD2 Taq1* is 40% in Koreans. The number of individuals with the *C957T* T allele was 14 (8.3%); they were labeled as “*C957T* risk allele carriers”, and others *(n* = 154, 91.7%) were labeled as “*C957T* nonrisk allele carriers”. The number of individuals with the *DRD2 Taq1* A1 allele was 119 (70.8%); they were labeled as “*Taq1* risk allele carriers”, and others (*n* = 49, 29.2%) were labeled as “*Taq1* nonrisk allele carriers”. The number of individuals who had both the *C957T* T allele and the *Taq1* allele was 10 (6.3%); they were labeled as “combined DRD2 risk allele carriers”, and others (*n* = 158, 93.8%) were labeled as “combined DRD2 nonrisk allele carriers”.

A Hardy–Weinberg equilibrium analysis of *DRD2 C957T* showed no deviation from HWE in the genotype distribution among all study participants; *x*^2^(1) = 0.071, *p* = 0.79. *DRD2 Taq1* also showed no deviation from HWE in the genotype distribution among all study participants; *x*^2^(2) = 1.38, *p* = 0.50.

### 3.2. Power Analysis

We performed a power analysis by using the Quanto program version 1.2.4 to detect whether our sample size had sufficient statistical power for the GxE model. We could not analyze the power analysis for the GxGxE model, because the complex model is not available in the Quanto program. We based expected effect sizes (*R*^2^) of the interaction between a DRD2 and stressful environments on the severity of alcoholism (*R*^2^ = 0.07) [23], main effect of the DRD2 genotype on problematic gaming (*R*^2^ = 0.11) [15], and main effect of interpersonal stressful environments on problematic gaming (*R*^2^ = 0.09) [21]. As a result, the sample size necessary for the power of 0.95 was 151. The result indicated the sample size of 168 subjects in this study had sufficient power to investigate our study aims of analyzing GxE effects. However, in the only study [15] that reported the main effect size of DRD2 on problematic gaming, the effect size (*R*^2^ = 0.11) seems unusually high for genetic main effect. Thus, although the outcome is not problematic gaming, we ran additional power analysis, using the more realistic main effect of DRD2 on gambling (*R*^2^ = 0.03) [56]. The result showed that the sample size necessary for the power of 0.80 was 157, and the size necessary for the power of 0.95 was 241.

### 3.3. Interactional Effects of DRD2 C957T or DRD2 Taq1 and Interpersonal Stress on Problematic Gaming

As shown in the left panel of Figure 1, *DRD2 C957T* had a significant GxE interaction effect on problematic gaming (*b* = 127.15, *β* = 0.26, *p* = 0.02). The main effect of *DRD2 C957T* (T allele) on problematic gaming was significant (*b* = 19.58, *β* = 0.24, *p* = 0.04), but the main effect of interpersonal stress (*b* = 5.89, *β* = 0.10, *p* = 0.12) was not significant. However, as shown in the right panel of Figure 1, *DRD2 Taq1* had no significant GxE interaction effect on problematic gaming (*b* = 5.87, *β* = 0.09, *p* = 0.49). The main effects of *DRD2 Taq1* (A1 allele; *b* = 0.14, *β* = 0.003, *p* = 0.96) and interpersonal stress (*b* = 1.71, *β* = 0.03, *p* = 0.83) on problematic gaming were not significant.

### 3.4. Mediated Moderating Effect of DRD2 C957T on Problematic Gaming

Because the GxE effect was significant only in the *DRD2 C957T*, the mediated moderating model was analyzed only using the *DRD2 C957T* polymorphism. As shown in Figure 2, avoidant coping did not significantly mediate the GxE effects on problematic gaming (*b* = 27.27, SE = 29.67, *p* = 0.36, 95% CI (40.55 to 160.88)). Avoidant coping positively predicted problematic gaming (*b* = 1.26, *β* = 0.20, *p* = 0.005), but interpersonal stress to avoidant coping was not significant (*b* = 21.63, *β* = 0.15, *p* = 0.28). The direct path from the GxE on problematic gaming was significant (*b* = 99.88, *β* = 0.12, *p* = 0.001).

### 3.5. Interactional Effects of Combined DRD2 Alleles and Interpersonal Stress on Problematic Gaming

As shown in Figure 3, the results demonstrated a significant GxGxE interaction effect on problematic gaming (*b* = 137.95, *β* = 0.24, *p* = 0.04). The main effect of combined DRD2 polymorphism on problematic gaming was significant (*b* = 22.35, *β* = 0.23, *p* = 0.05), but the main effect of interpersonal stress on problematic gaming (*b* = 5.99, *β* = 0.10, *p* = 0.12) was not significant.

To probe the significant GxGxE interaction from the path analysis, we conducted multigroup analysis as a function of the combined DRD2 risk allele carriers versus non-carriers. Because college students experience higher levels of interpersonal stress, DRD2 risk allele carriers were more likely to engage in problematic gaming (*b* = 151.70, *β* = 0.72, *p* < 0.001), whereas no significant association was found among non-carriers (*b* = 5.95, *β* = 0.10, *p* = 0.13). Figure 4 shows the estimated means of problematic gaming as a function of the DRD2 genotype (carriers of both *C957T* T allele and *Taq1* A1 allele versus non-carriers) and interpersonal stress (upper 50% versus lower 50%), illustrating the greater vulnerability to interpersonal stress among carriers as compared with non-carriers.

### 3.6. Mediated Moderating Effect of Combined DRD2 Alleles on Problematic Gaming

As shown in Figure 5, the results demonstrated that avoidant coping significantly mediated the GxGxE effects on problematic gaming (*b* = 51.79, *β* = 0.05, *p* = 0.02, 95% CI (14.05 to 103.74)). The interaction between DRD2 risk alleles and interpersonal stress positively predicted avoidant coping (*b* = 42.03, *β* = 0.25, *p* < 0.001), and, in turn, avoidant coping positively predicted problematic gaming (*b* = 1.23, *β* = 0.20, *p* = 0.01). In addition, the direct path from the GxGxE on problematic gaming was significant (*b* = 86.43, *β* = 0.09, *p* = 0.01). These results indicated that, when experiencing interpersonal stress, individuals with DRD2 risk alleles (*C957T* T allele and *Taq1* A1 allele) are likely to use avoidant coping strategies, which increases the severity of problematic gaming. As shown in Table 3, avoidant coping significantly mediated the effects of interpersonal stress on problematic gaming among DRD2 risk allele carriers (indirect effect = 52.45, SE = 22.72, *p* = 0.02, 95% CI (14.21 to 104.91)). However, the mediating effect of avoidant coping was insignificant among non-carriers (indirect effect = 0.66, SE = 0.79, *p* = 0.41, 95% CI (−0.58 to 2.68)).

### 3.7. Sensitivity Analyses

We conducted two sets of analyses that examine the GxGxE interaction, using other genotype categorizations: (a) identifying the C allele (not T allele) for the *C957T* polymorphism as a risky allele and (b) three-group categorizations (2 = carrying both *Taq1* A1 allele and *C957T* T allele, 1 = carrying either *Taq1* A1 allele or *C957T* T allele, 0 = not carrying any of *Taq1* A1 allele or *C957T* T allele). First, after the C allele of the *C957T* polymorphism was considered a risky allele, 109 (64.9%) participants carrying both the C allele of *DRD2 C957T* and the A1 allele of *DRD2 Taq1* were placed in the risk group. Genetic risk involving both the C allele for *C957T* polymorphism and A1 of *DRD2 Taq1* has no significant interaction effect when combined with interpersonal stress on problematic gaming (*b* = 4.25, *β* = 0.06, *p* = 0.62). Second, we used the abovementioned three group categorization: 10 (6.0%) individuals carried both the *Taq1* A1 allele and the *C957T* T allele, 113 (67.3%) carried either the *Taq1* A1 allele or the *C957T* T allele, and 45 (26.8%) carried neither the *Taq1* A1 allele nor the *C957T* T allele. The GxGxE interaction effects using three-group categorization also did not show a significant effect on problematic gaming (*b* = 9.71, *β* = 0.15, *p* = 0.25).

## 4. Discussion

In this study, we investigated the effect of DRD2 polymorphisms and interpersonal stress on problematic gaming and determined whether avoidant coping strategies mediate and explain the GxE and GxGxE effect mechanism. The results of the moderation and multigroup analyses showed that interpersonal stress is more likely to increase problematic gaming in individuals carrying both the *DRD2 C957T* T allele and the *Taq1* A1 allele than in non-carriers. The results of the mediated moderation analysis showed that individuals carrying the *DRD2 C957T* T allele and the *DRD2 Taq1* A1 allele tended to use avoidant coping strategies to address interpersonal stress, leading to more severe problematic gaming than in non-carriers.

Our findings demonstrate that a hypodopaminergic dysfunctional state combined with elevated interpersonal stress has a synergistic effect on vulnerability to problematic gaming. The significant GxGxE interaction effect is consistent with the diathesis-stress model, which asserts that an individual’s negative psychological or physical state may be the result of an innate predisposition (e.g., genetic factors) to react vulnerably to negative environments, such as stressful situations [57]. Consistent with this model, our results demonstrate that a stressful environment interacts with DRD2 polymorphisms to increase vulnerability to problematic gaming. Our results also can be interpreted as the loss of functional autonomy of the mesolimbic dopamine-dependent seeking system [58]. Decreased dopaminergic function may elevate individuals’ emotional drive to seek addictive stimuli. Thus, those with decreased dopamine function are likely to lose the ability to suppress addictive behaviors and have an emotion command system [59]. The *DRD2 C957T* T allele and the *DRD2 Taq1* A1 allele, which both have been associated with a low functioning of the D2 receptor, were found to increase vulnerability to problematic gaming. Previous neurobiological studies have found that the *DRD2 C957T* T allele reduces mRNA translation efficiency and attenuates the dopamine-induced upregulation of DRD2 expression [12,13]. In addition, studies using positron emission tomography have found that those subjects carrying the *DRD2 Taq1* A1 allele exhibited lower striatal D2 receptor solubility and lower putamen D2 receptor-binding potentials than subjects carrying the A2 allele [60,61]. The negative cooperative influence of *C957T* and *Taq1* may have further exacerbated the low functioning of the D2 receptor; this may be possibly because the *C957T* variant affects receptor affinity, whereas the TaqIA A1 polymorphism affects the Bmax receptor [13,62]. 

When we separately analyzed the GxE interactions with two DRD2 polymorphisms, the interaction effect of *DRD2 C957T* with interpersonal stress on problematic gaming was significant, but the interaction effect of *DRD2 Taq1* was not significant. Despite relatively consistent evidence from neurobiological studies demonstrating that the decreased D2 receptor function was observed in problematic gamers [63,64], inconsistency exists in the association of DRD2 single polymorphism with problematic gaming [15,65]. This may be because more than one genetic polymorphism influences problematic gaming: perhaps the combined presence of the T allele of *DRD2 C957T* involved in reduced mRNA translation [12] and the A1 allele of *DRD2 Taq1* involved in reduced receptor affinity [13] increased vulnerability to avoidant coping more so than in subjects carrying a single gene. In line with our findings, the combined genetic risk of *DRD2 C957T* and the *DRD2 Taq1* polymorphisms significantly affected both alcohol [18,19] and nicotine dependence [51].

The mediated moderation model using combined DRD2 allele risks demonstrates that avoidant coping works as a mechanism to increase the severity of problematic gaming. This result indicates that a possible reason for excessive gaming among risk allele carriers may be their relatively greater tendency to avoid directly addressing interpersonal problems. Persistent escapism can develop into addiction symptoms, as coping behaviors to avoid real-life problems repeat [66,67]. Individuals who use games with a motivation to escape from life’s problems were found to experience increased negative outcomes, such as depression and stress [68,69,70]. That is, responding to stress with avoidant coping strategies eventually increases negative emotions, which ultimately leads to a vicious cycle of avoidance [31]. Neurobiological studies have shown that the dopaminergic system is activated in a stressful situation to eliminate stressors and induce motivation to cope with stress through the activation of dopamine D2 receptors [71,72]. If the dopamine does not sufficiently perform its neurochemical role, individuals may not feel sufficiently competent to manage their stress, opting instead for passive or avoidant coping mechanisms [73]. Therefore, the *DRD2 C957T* T allele and the *DRD2 Taq1* A1 allele can cause dopaminergic system dysfunctions in stress coping and response, which lead to lower tolerance of negative experiences.

The current study’s findings have potential clinical implications for prevention and intervention efforts to curtail problematic gaming among young adults. In general, GxE findings allow us to identify “high-risk” groups of individuals who are relatively more vulnerable to certain social environmental effects. Although individuals’ genotypes cannot be changed, GxE findings can help us design targeted prevention or intervention strategies for populations at risk of problematic gaming. Findings from this study suggest that interpersonal stressors need to be specifically addressed in intervention or prevention programs for DRD2 risk allele carriers. For college students, in particular, who carry both the *DRD2 C957T* T allele and the *DRD2 Taq1* A1 allele, prevention and intervention strategies need to include monitoring individuals’ own assessment of the controllability of interpersonal stressors and the development of problem-solving strategies in interpersonal relationships. A previous study demonstrated that a family prevention program targeting overall parenting competence and control effectively delayed risk behavior initiation in the 5-hydroxy tryptamine transporter-linked polymorphic region (5-HTTLPR) risk genotype carriers more than in non-carriers [74]. This finding highlights the promise of genetically informed intervention efforts to reduce addictive behaviors. A selective prevention and intervention approach for individuals with the high-risk genotype may be more efficient and effective than a universal approach targeting all adolescents regardless of present genetic and environmental risks.

The findings of this study should be interpreted in consideration of the following limitations. First, although the candidate genes of this study were included based on previous studies [15,16,17], other candidate genes related to dopamine functions may interplay with DRD2 genotypes in affecting problematic gaming. For example, the dopamine D4 receptor gene (DRD4) 48bp variable number of tandem repetition (VNTR) [16,75] and the dopamine transporter gene (DAT1) 40bp VNTR [76] have been associated with internet addiction. Therefore, because the effect of a single genetic variant on complex behavior such as problematic gaming is most likely minimal, examining the cumulative genetic effect of multiple dopamine-related genetic variants would identify a more comprehensive genetic profile. Second, there is a potential concern of false positives in our results because of the small minor allele frequency [77,78] within the study group. In our study, the frequency of carrying both the *DRD2 C957T* T allele and the *DRD2 Taq1* A1 allele was low at 6%. In many studies, single-nucleotide polymorphisms (SNPs) that had minor allele frequencies (MAF) of less than 5% or 10% were excluded [79,80,81]. However, some studies have asserted that discarding less than 5% of SNPs may interfere with the ability to detect rare disease-causing polymorphisms because SNPs with low MAF are more likely to be functional [82], and analyses using SNPs with minor allele frequencies (<5% or 1%) did not demonstrate high false positives [83]. Because of the controversies over minor allele frequency, our findings need to be replicated across several independent samples in future studies. Third, there are limitations on interpreting the causality of the mediated moderation effect model because our study used the cross-sectional study design. Therefore, it is necessary to confirm the causal direction of our mediated moderation model by using a longitudinal study design in future studies. Finally, it is necessary to further investigate how alternative splicing mechanisms involved in the dopaminergic control of the D2 receptor affect problematic gaming beyond genetic polymorphisms. The two alternative D2R mRNA splice variants, D2 receptor-long (D2L) and D2 receptor-short (D2S), have been found to play an important role in substance addiction [84] and addiction-like phenotypes [85]. Despite these potential limitations, however, this study contributes to the research on the G × G × E for problematic gaming as the first study to explore the effect of interpersonal stress and the interaction of DRD2 genetic factors on problematic gaming and investigate the mediating effect of avoidant coping strategies.

## Figures and Tables

**Figure 1 genes-13-00449-f001:**
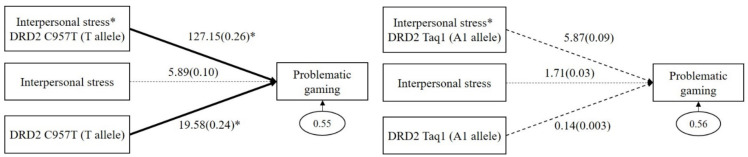
Interaction effect between *DRD2 C957T* and interpersonal stress on problematic gaming (in the left panel), and interaction effect between *DRD2 Taq1* and interpersonal stress on problematic gaming (in the right panel). Sex, impulsivity, depression, anxiety, and ADHD were controlled for in all analyses (paths not shown for simplicity). Unstandardized (outside parentheses) and standardized (inside parentheses) coefficients are shown. Solid lines represent statistically significant coefficients. Dashed lines represent statistically insignificant coefficients; * *p* < 0.05.

**Figure 2 genes-13-00449-f002:**
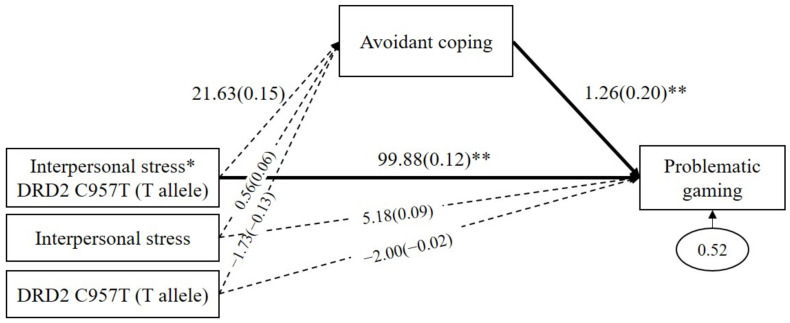
Mediated moderation model with avoidant coping as the mediator in the interaction effect of the *DRD2 C957T* and interpersonal stress on problematic gaming. Sex, impulsivity, depression, anxiety, and ADHD were controlled for in all analyses (paths not shown for simplicity). Unstandardized (outside parentheses) and standardized (inside parentheses) coefficients are shown. Solid lines represent statistically significant coefficients. Dashed lines represent statistically insignificant coefficients; * *p* < 0.05, ** *p* < 0.01.

**Figure 3 genes-13-00449-f003:**
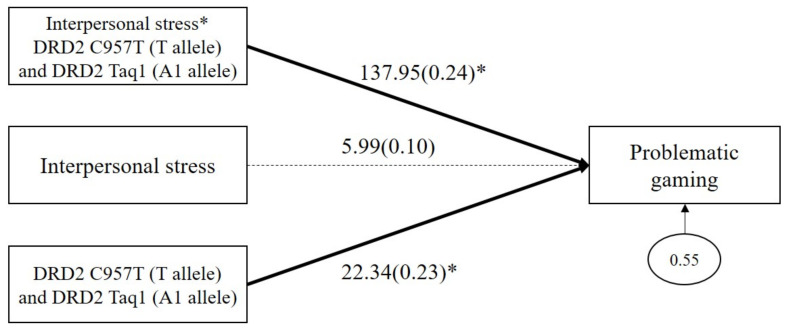
Interaction effect between combined DRD2 polymorphisms and interpersonal stress on problematic gaming. Sex, impulsivity, depression, anxiety, and ADHD were controlled for in all analyses (paths not shown for simplicity). Unstandardized (outside parentheses) and standardized (inside parentheses) coefficients are shown. Solid lines represent statistically significant coefficients. Dashed lines represent statistically insignificant coefficients; * *p* < 0.05.

**Figure 4 genes-13-00449-f004:**
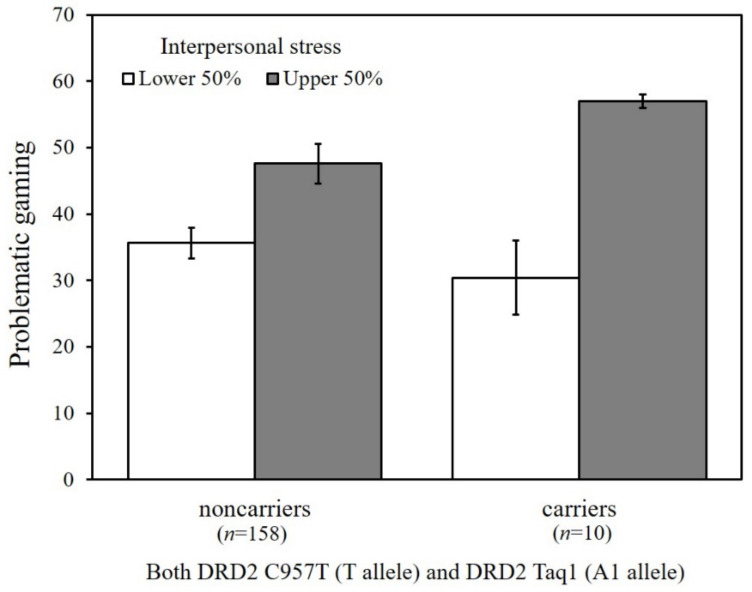
Estimated means of problematic gaming as a function of the *DRD2 C957T* and *Taq1* allele types and levels of interpersonal stress. Vertical bars represent the standard error below and above the mean scores.

**Figure 5 genes-13-00449-f005:**
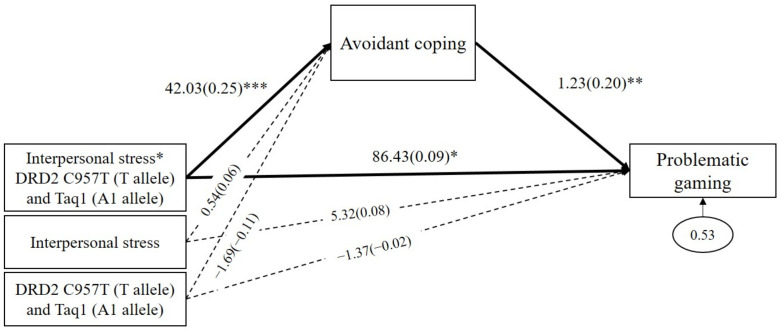
Mediated moderation model with avoidant coping as the mediator in the interaction effect of combined DRD2 polymorphisms and interpersonal stress on problematic gaming. Sex, impulsivity, depression, anxiety, and ADHD were controlled for in all analyses (paths not shown for simplicity). Unstandardized (outside parentheses) and standardized (inside parentheses) coefficients are shown. Solid lines represent statistically significant coefficients. Dashed lines represent statistically insignificant coefficients; * *p* < 0.05, ** *p* < 0.01, and *** *p* < 0.001.

**Table 1 genes-13-00449-t001:** Means, standard deviations, and correlation of study variables.

Variables (Range)	1	2	3	4	5	6	7	8	9	10	11	12
**1. *DRD2 C957T* risk allele**	-											
**2. *DRD2 Taq1* risk allele**	0.004	-										
**3. Combined risk allele**	0.83 ***	0.16 *	-									
**4. Male**	0.09	0.03	0.08	-								
**5. Age (19–33)**	0.02	−0.05	−0.05	0.23 **	-							
**6. Problematic gaming (1–105)**	−0.07	−0.02	−0.03	−0.30 ***	−0.02	-						
**7. Interpersonal stress (0–15)**	−0.11	0.01	−0.10	−0.03	0.08	0.29 ***	-					
**8. Avoidant coping (6–24)**	−0.10	0.03	−0.02	−0.21 **	−0.03	0.49 ***	0.19 *	-				
**9. Impulsivity (22–69)**	−0.003	−0.04	0.05	−0.25 **	0.08	0.47 ***	0.20 **	0.50 ***	-			
**10. Depression (0–25)**	−0.11	−0.02	−0.08	−0.26 **	−0.07	0.57 ***	0.31 ***	0.42 ***	0.50 ***	-		
**11. Anxiety (0–21)**	−0.10	−0.01	−0.03	−0.15 *	0.15	0.50 ***	0.40 ***	0.28 ***	0.42 ***	0.71 ***	-	
**12. ADHD (0–6)**	−0.13	−0.05	−0.09	−0.36 ***	−0.05	0.53 ***	0.20 ***	0.41 ***	0.59 ***	0.47 ***	0.40 ***	-
**Mean**	8.3% ^a^	70.6% ^a^	6.0% ^a^	63.1% ^a^	22.09	39.67	0.93	12.37	42.88	6.36	4.83	2.81
**Standard deviation**	-	-	-	-	2.35	23.05	1.91	3.73	8.64	5.16	5.01	1.68
**Skewness**	3.04	−0.92	3.76	−0.55	0.76	0.44	3.68	0.27	0.23	1.22	1.29	−0.17
**Kurtosis**	7.34	−1.16	12.26	−1.72	1.37	−0.44	19.37	−0.69	0.17	1.30	1.05	−1.07

Note: Correlations between dichotomous variables (e.g., *DRD2 C957T* risk allele, *DRD2 Taq1* risk allele, combined DRD2 risk allele, and sex) and other variables were reported as Spearman’s correlation coefficients, and correlations between continuous variables were reported as Pearson’s correlation coefficients; * *p* < 0.05, ** *p* < 0.01, and *** *p* < 0.001. ^a^ Sex was presented as a as a percentile.

**Table 2 genes-13-00449-t002:** Mean and percentages of study variables as DRD2 allele types.

**Variables (Range)**	***DRD2 C957T* Risk Allele Carriers** **(*n* = 14)**	***DRD2 C957T* Nonrisk** **Allele Carriers** **(*n* = 154)**	**Test Statistics** **Differences**
	*M*(*SD*)	*M*(*SD*)	
Male	80%	62%	*x*^2^(1) = 1.57
Age (19–33)	22.29(2.52)	22.07(2.35)	*t*(166) = 0.33
Problematic gaming (1–105)	34.21(21.47)	40.17(23.20)	*t*(166) = −0.93
Interpersonal stress (0–15)	0.21(0.43)	1.00(1.98)	*t*(85.86) = −4.01 *^,a^
Avoidant coping (6–22)	11.21(4.37)	12.47(3.66)	*t*(166) = −1.21
Impulsivity (22–69)	42.50(8.79)	42.92(8.66)	*t*(166) = −0.17
Depression (0–25)	4.64(4.29)	6.52(5.21)	*t*(166) = −1.31
Anxiety (0–21)	2.57(1.95)	5.03(5.16)	*t*(33.63) = −3.69 *^,a^
ADHD (0–6)	2.14(1.17)	2.87(2.87)	*t*(18.48) = −2.13 *^,a^
**Variables (Range)**	***DRD2 Taq1* Risk Allele Carriers** (***n* = 119)**	***DRD2 Taq1* Nonrisk** **Allele Carriers** **(*n* = 49)**	**Test Statistics** **Differences**
	*M*(*SD*)	*M*(*SD*)	
Male	80%	62%	*x*^2^(1) = 0.10
Age (19–33)	22.03(2.41)	22.22(2.23)	*t*(166) = −0.48
Problematic gaming (1–105)	39.39(23.30)	40.37(22.66)	*t*(166) = −0.25
Interpersonal stress (0–15)	0.96(2.00)	0.88(1.68)	*t*(166) = 0.25
Avoidant coping (6–22)	12.49(3.89)	12.08(3.32)	*t*(166) = 0.64
Impulsivity (22–69)	42.66(9.03)	43.43(7.67)	*t*(166) = −0.53
Depression (0–25)	6.36(5.34)	6.37(4.73)	*t*(166) = −0.01
Anxiety (0–21)	4.91(5.16)	4.63(4.68)	*t*(18.85) = 0.32
ADHD (0–6)	2.74(1.75)	2.98(1.48)	*t*(166) = −0.84
**Variables (Range)**	**Combined DRD2 Risk Allele Carriers** **(*n* = 10)**	**Combined DRD2 NonRisk Allele Carriers** **(*n* = 158)**	**Test Statistics** **Differences**
	*M*(*SD*)	*M*(*SD*)	
Male	80%	62%	*x*^2^(1) = 1.31
Age (19–33)	21.60(2.01)	22.12(2.38)	*t*(166) = −0.67
Problematic gaming (1–105)	35.70(17.89)	39.92(23.37)	*t*(166) = −0.56
Interpersonal stress (0–15)	0.20(0.42)	0.98(1.96)	*t*(166) = −1.26
Avoidant coping (6–22)	12.10(4.75)	12.39(3.67)	*t*(166) = −0.24
Impulsivity (22–69)	43.80(7.76)	42.82(8.71)	*t*(166) = 0.35
Depression (0–25)	4.50(3.34)	6.48(5.23)	*t*(166) = −1.18
Anxiety (0–21)	3.10(1.91)	4.94(5.13)	*t*(18.85) = −2.52 *^,a^
ADHD (0–6)	2.30(0.95)	2.84(1.71)	*t*(13.05) = −1.65 ^a^

Note: Combined DRD2 risk allele carriers refer to individuals carrying both the T allele of *DRD2 C957T* and the A1 allele of *DRD2 Taq1*. Combined DRD2 nonrisk allele carriers are those carrying only one or neither of the T allele of *DRD2 C957T* and the A1 allele of *DRD2 Taq1*; * *p* < 0.05. ^a^ Because equal variance was not assumed, Welch’s statistical method was used.

**Table 3 genes-13-00449-t003:** Conditional indirect effect.

	Conditional Indirect Effect by Combined DRD2 Allele Type			
	Estimate	SE	Bootstrap Lower 95% CI	Bootstrap Upper 95% CI
Combined DRD2 nonrisk allele carriers	0.66	0.79	−0.58	2.68
Combined DRD2 risk allele carriers	52.45 *	22.73	14.21	104.91

Note: Combined DRD2 risk allele carriers refer to individuals carrying both the T allele of *DRD2 C957T* and the A1 allele of *DRD2 Taq1*. Combined DRD2 nonrisk allele carriers are those carrying only one or neither of the T allele of *DRD2 C957T* and the A1 allele of *DRD2 Taq1*; * *p* < 0.05.

## Data Availability

Not applicable.

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
