# Peer review of "Interaction Effects of DRD2 Genetic Polymorphism and Interpersonal Stress on Problematic Gaming in College Students"

_genes, 2022, doi:10.3390/genes13030449_

Round 1

Reviewer 1 Report

It is a very interesting study exploring the possible mediating effect of avoidant coping between interpersonal stress, DRD2 gene polymorphisms and gaming. The analysis itself is very complex, the manuscript is well written, the used methods are described in detail.

However, this complexity might also be the biggest weakness of the study. The study investigates the relationship of two genetic markers, two psychological factors with also adding five covariates. On a sample of 168 college students. This sample size is rather small, especially in case of the group of the two risk alleles where n = 10. Although the authors included a power analysis, it seems to have some problems. For example, the effect size of main effect of DRD2 genotype on problematic gambling was set as 0.11 which seems high for genetic main effects, and extremely high for and effect of a combined genetic risk allele of two genetic polymorphisms. Furthermore, the authors call this study a GxE interaction analysis, while the truth is that it is a GxGxE analysis at least – but rather a GxGxExE taking into consideration the analyzed moderating factor (+ further five factors were included as covariates).

Also, preregistering the study aim would have strengthened the present analysis a lot. When many variants are measured in a study, preregistration of the study aims and the hypothesis is very beneficial, since it is a proof that the analysis was planned previously and not after many explorative analyses of the data.

Further comments:

  1. Escapism is a very important psychological factor in behavioral addictions and it does share some commonalities with avoidant coping. I would suggest to include its literature in the Introduction/Discussion section.
  2. Results the association analysis of the two DRD2 polymorphisms separately should be included in the manuscript.
  3. Interpersonal stress scale mean score was reported to be 0.93 with a Std of 1.91 on the present sample. Since the scale ranges between 0-15 this mean score seems to be rather low, maybe showing a floor effect. Maybe the floor effect and the low number of participants is the reason why the difference in interpersonal stress in Table 2 is ns.
  4. Please avoid writing p = 0.00. like in page 5 line 203. The level of significance is never 0.00. Write p <0.01

Reviewer 2 Report

The study of Kim and colleagues examined the interaction effects of DRD2 polymorphisms and interpersonal stress on problematic gaming and the mediating influence of an avoidant coping to reveal the mechanism of the GxE process. The article is well written, and the research appears to be scientifically sound. I have only a single concern regarding the substantial difference between the number of carriers (n=158) and noncarriers (n=10) in the multigroup analysis depicted in Figure 2. Is the multigroup analysis robust to such a significant difference? Please clarify this point and, if needed, apply the appropriate statistical correction. 
Moreover, I'd like to suggest the author discuss the existence of another layer of complexity beyond D2R polymorphism. Indeed, alternative splicing can impact D2R function beyond genetic polymorphism. For example, genetic variability in the proportion of the two alternative D2R mRNA splice variants, D2R-long (D2L) and D2R-short (D2S), influences corticostriatal functioning. In particular, Colelli et al. (2010) have shown that differences in tissue-specific D2R splicing influence individual variability in addiction-like phenotypes (https://doi.org/10.1111/j.1601-183x.2010.00604.x).
Furthermore, the authors would further improve the article's breadth by discussing their results in the light of Panksepp's affective neuroscience theory. For example, they could interpret their results in terms of loss of "functional autonomy" of the mesolimbic dopamine-dependent SEEKING system (https://doi.org/10.3389/fnhum.2021.635932).

Reviewer 3 Report

Page 8 lines 315-316 :”… Our findings showed that a hypodopaminergic dysfunctional state combined with the elevated stress levels from social defeat has a synergistic effect on vulnerability to problematic gaming. “  There is no mention in the paper of a relationship between any of the stress measures evaluated and ‘social defeat’; please discuss evidence in support of this relationship or do not refer to social defeat.

Page 9 lines 322-323: “…The DRD2 C957T T allele and the DRD2 Taq1 A1 allele, which have been associated with decreased dopamine synthesis…”  and lines 329-330 “The negative cooperative influence of C957T and Taq1 may have further exacerbated the insufficient dopamine release”. If the meaning of these phrases is that the physiological target of the two variants is dopamine availability, this is not true. The two variants could cooperate to reduce D2 dopamine receptors functioning in the striatum, possibly because the C957T variant affects receptor affinity while the TaqIA A1 polymorphism affects the receptor Bmax (DOI: 10.1002/syn.21916; your references 13,14,55, 56). So reduced dopamine transmission does not depend on reduced neurotransmitter synthesis or release but low receptors functioning.  Please, correct.
